# The Real Time Geometric Effect of a Lordotic Curve-Controlled Spinal Traction Device: A Randomized Cross Over Study

**DOI:** 10.3390/healthcare9020125

**Published:** 2021-01-27

**Authors:** Chang-Hyung Lee, Sung Jin Heo, So Hyun Park

**Affiliations:** 1Department of Rehabilitation Medicine, Pusan National University School of Medicine and Research Institute for Convergence of Biomedical Science and Technology, Pusan National University Yangsan Hospital, Daegu 50612, Gyeongsangnam-do, Korea; whitegusdl@gmail.com; 2Department of Physical Therapy, Youngsan University, Gyeongsan 50510, Gyeongnam-do, Korea; ptpsh@ysu.ac.kr

**Keywords:** lordotic curve-controlled traction device (LCCT), lordotic curve, intervertebral disc space

## Abstract

*Background:* A standard spinal traction (ST) device was designed to straighten the spine without considering physiological lumbar lordosis. Using lordotic curve-controlled traction (LCCT), which maintains the lordotic curve during traction, the traction force would be applied to the posterior spinal structure effectively. Thus, the purpose of our study was to evaluate real-time biomechanical changes while applying the LCCT and ST. *Methods:* In this study, 40 subjects with mild non-radicular low back pain (LBP) were included. The participants underwent LCCT and ST in random order. Anterior and posterior intervertebral distance, ratios of anterior/posterior intervertebral distance (A/P ratio), and lordotic angles of intervertebral bodies (L2~L5) were measured by radiography. *Results:* Mean intervertebral distances were greater during LCCT than those measured prior to applying traction (*p* < 0.05). Mean A/P ratio was also significantly greater during LCCT than during ST or initially (*p* < 0.05). In particular, for the L4/5 intervertebral segment, which is responsible for most of the lordotic curve, mean LCCT angle was similar to mean lordotic angle in the standing position (10.9°). *Conclusions:* Based on measurements of radiologic geometrical changes with real-time clinical setting, the newly developed LCCT appears to be a useful traction device for increasing intervertebral disc spaces by maintaining lordotic curves.

## 1. Introduction 

Spinal traction (ST) is a form of decompression therapy which was designed to relieve pressure on the spine and improve spinal alignment. Among the many beneficial effects of ST, its main aim is to provide pain relief, help achieve spinal alignment, and relieve pressure on nerves, particularly in patients with disc disease [1]. 

Traction devices increase the intervertebral distance in patients with disc diseases. However, despite the many benefits of ST, mixed results have been reported for herniated intervertebral discs (HIVDs) [2,3,4,5,6]. Unfortunately, not many high-quality randomized, controlled studies have been performed on ST therapy [2,4]. For example, some reported studies have lacked a randomized control group [2,3], been conducted retrospectively [7,8,9,10], or had small sample sizes [7,8,11]. Furthermore, various traction techniques and protocols have been used, and thus, the reported results vary.

In addition, the decompressing effects of traction therapy are unclear. Clarke et al. [4] asserted that this was because the effects of traction are difficult to scientifically determine in clinical cases for the following reasons: (1) difficulties in setting up a pre-traction conditions condition, (2) difficulties in conducting blind tests with a mechanical traction stress, (3) different education levels of the patients, (4) difference in the understanding of the mechanisms of diseases, and (5) different possible causes of the same symptoms or alternatively, different symptoms with the same possible cause.

Despite the theoretically favorable effects of ST devices, clinical reliance on traction therapy for the treatment of lumbar disc disease is low. Improper pressure loading on disc structures during traction is a possible explanation for this lack of clinical efficacy. In the supine position, the lordotic curve decreases due to vertical pressure on the lumbar curve, which is more decreased in patients with low back pain [12]. When traction force is applied in the supine position, it primarily straightens the natural lordotic curve without unduly decompressing the intervertebral discs. Thus, rather than decompressing the intervertebral discs, traditional traction decreases the lordotic curve [13]. Under ST, posterior spinal structures, such as facet joints and posterior longitudinal and interspinous ligaments, are elongated more than anterior spinal structures [13]. When traction pressure is applied to the spine in the supine position using a standard traction device, the lordotic curve decreases at the expense of equal distraction of the whole spinal structure, and consequently, pain develops.

Thus, we considered that if the traction force could be applied to the vertebrae while maintaining the lordotic curve, the force would be distributed equally to the anterior and posterior parts of the spinal structure. Subsequent development resulted in a lumbar lordotic curve-controlled traction device (LCCT) targeting the L4/5 intervertebral disc space.

To our knowledge, no study has evaluated the clinical efficacy of LCCT by measuring its angle and distance in real-time clinical settings. In the present study, we evaluated intervertebral disc space changes during LCCT visualized using radiography and compared the results these to those obtained during standard traction (ST).

## 2. Methods

### 2.1. Subjects

This prospective, randomized, case-controlled crossover study was conducted at our clinic. All participants were recruited from our clinic in Pusan National University Yangsan Hospital, Korea, and each participant provided informed consent before study commencement. Forty participants with mild, non-radicular low back pain (LBP) treated from April to November 2016 were included. Participants having LBP symptoms since <3 months, systemic inflammation, a spinal structural abnormality (including a spinal fracture history), multiple disc protrusions, spondylolisthesis, or any other structural deformity were excluded. During the initial, acute stage of low back pain, muscle guarding or spasm occurs, which exacerbate the spinal asymmetry and nerve root irritation. Thus, unnecessary muscle guarding could interfere with the real-time effect of traction in this study. The study was approved by the local and the central ethical committees at our hospital (03-2016-008).

### 2.2. Lordotic Curve-Controlled Traction Device (LCCT) versus a Standard Traction Device (ST) 

The lordotic curve-controlled traction device (LCCT) was designed to maintain the natural lordotic curve by supporting the lumbar curve at the L4/5 intervertebral disc space. We used a commercial product (Kinetrac-9900, Hanmed Co., Gimhae, Korea) with some modifications to set the clinical setting. Initially, a magnetic marker was attached on the L4/5 intervertebral disc space by physical palpation, and an automated tracking system, which also maintained the lumbar lordotic curve by elevating L4/5, used this marker during ST. During LCCT, the height of the elevated form of lumbar lordotic curve support was increased to the most comfortable point for participants. Conversely, the ST method was applied as usual without an aim to maintain the lumbar lordotic curve during traction. In both procedures, patients lay in supine position on the traction table with knee supports placed under each knee. Traction was applied and gradually increased to the maximal tolerable level or until the force reached one third of the patient’s weight. LCCT and ST traction methods are summarized in Figure 1.

### 2.3. Radiologic Evaluation of Intervertebral Disc Space Changes: Measurements of Intervertebral Angle and Distance Changes during Traction 

Using C-arm equipment (Siemens, German), initially, we obtained images in lateral views in the standing and supine position. In standing position, radiograph was taken in comfortable standing posture with cross-armed position and keeping feet and legs shoulder-width apart. The patients were assigned to traction method ST or LCCT in random order. The random allocation was decided by tossing a coin. When the head side comes out, the subject was assigned to ST treatment first and the other, adversely. The first intervention was applied for 10 min and subjects had a washout period of 20 min between each intervention. After this, the patients were assigned to the other traction method in a crossover study design.

In a supine position, radiological imaging was performed before and after 10 min during conducting each traction device. Real-time shooting was performed during both traction types. The obtained images were sent to the Picture Archiving and Communication System (PACS, Infinitt healthcare Co., Ltd., Seoul, South Korea) and the images were analyzed using cobb angle and distance measurement in the PACS system.

The following items were measured by one blinded specialist in radiology: intervertebral disc angle of all segments, and intervertebral disc distances of anterior and posterior side (Figure 2). Intervertebral disc distance was defined as the distance between the inferior endplate of the upper vertebra and superior endplate of the opposing lower vertebra. Intervertebral disc angle was defined as the angle between the inferior endplate of a superior vertebral body and the superior endplate of the inferior vertebral body in lateral view.

## 3. Statistics

Sample size analysis showed that for a two-sided level of significance of 0.05 and an interclass correlation of 0.8, at least 37 participants were required [14], and thus, 40 patients were enrolled to cope with potential losses. To determine differences between mean L4/5 anterior/posterior distance ratios during initial, LCCT, and ST conditions, one-way analysis of variance (ANOVA) was used. Tukey’s post hoc test was used to investigate the differences between all groups. The analysis was performed using IBM SPSS Statistics for Windows (version 22.0). All study data were normally distributed, and statistical significance was accepted for *p*-values < 0.05.

## 4. Results

Patient demographic data are summarized in Table 1. Mean patient age was 40.3 ± 14.91 years (13 males, 27 females).

Mean intervertebral distances during initial, LCCT, and ST conditions for L2/3 were 21.61, 24.92, and 22.71 mm, for L3/4 were 23.62, 27.52, and 26.04 mm, and for L4/5 were 26.26, 29.82, and 27.49 mm respectively, and the difference between the initial and LCCT conditions was significant (*p* < 0.05) (Figure 3A). Mean intervertebral distances of posterior sides in lateral view for initial, LCCT, and ST conditions for L2/3 were 16.03, 17.94, and 16.78 mm, for L3/4 were 16.18, 19.20, and 18.20 mm, and for L4/5 were 16.69, 18.82, and 18.65 mm respectively, and the difference between initial and LCCT conditions was significant (*p* < 0.05) (Figure 3B). Distances between anterior and posterior sides were used to calculate anterior/posterior ratios (A/P ratio), and results under initial, LCCT, and ST conditions for L2/3 were 1.35, 1.35, and 1.38, for L3/4 were 1.46, 1.43, and 1.44, and for L4/5 were 1.59, 1.44, and 1.58, respectively (Figure 3C). A/P ratio was greatest at L4/5, where the lordotic curve is most pronounced (Figure 3C and Figure 4A). At the L4/5 level, A/P ratios were significantly different under initial, LCCT, and ST conditions (*p* < 0.05) and between initial and ST (*p* < 0.05) and LCCT and ST (*p* < 0.05) conditions. However, L4/5 A/P ratios under initial and LCCT conditions were similar (*p* > 0.05) (Figure 3C).

The initial and LCCT angles were greater than ST angles in all intervertebral segments. LCCT angles at individual discs were slightly but significantly greater than ST angles for all segments (*p* < 0.001). In fact, ST angles were smallest for all intervertebral segments (*p* < 0.001) (Figure 4A). In particular, for L4/5 intervertebral segments, the LCCT angles were similar to intervertebral discs in the standing position (10.9°) [15]. Thus, it appears that even when supine, LCCT maintains the lordotic curve at the average value in the standing position (10.5°) (Figure 4B).

During the study, one participant complained of back discomfort after ST, no such compliant was made after LCCT, and no study-related complication occurred.

## 5. Discussion

We focused on lordotic curve changes caused during traction depending on the traction method. We invented LCCT, a new method of traction which could maintain the physiological lordotic curve during traction therapy and analyzed the effects of LCCT on intervertebral disc distances and intervertebral disc angle of lumbar vertebral spines using radiographic imaging in a real-time setting. 

During ST, the force exerted decreases the lordotic curve and has a greater effect on the posterior structure and overstretching of the posterior structure could cause pain and muscle spasms. Furthermore, elongation of the posterior vertebral column could damage facet joints and soft tissue structures without having much effect on discs [13]. In addition, the natural lordotic curve decreases during traction in the supine position than the standing position. These biomechanical changes with ST could decrease the traction effects.

We suggest the developed LCCT device can maintain a natural lordotic curve during traction by providing lumbar support, which means traction force is evenly distributed, increases negative pressure, and produces favorable results. In addition, the provision of this support decreases unnecessary pain and muscle spasms, and thus, allows greater forces to be applied.

In the present study, the LCCT was found to differ from ST in several ways. When the same amount of force was applied, LCCT showed greater distance increases than ST, which was possibly due to fewer muscle spasms, as they tend to occur when too much force is applied to the posterior column. Thus, we suppose that as the lordotic curve is preserved during LCCT, applied force is better distributed than during ST. Park et al. [13], who conducted finite element analysis, reported similar trends, and concluded fibers of the annulus fibrosus in the posterior region and intertransverse and posterior longitudinal ligaments experience greater stress during standard traction, and that these stresses are reduced during traction when the lumbar lordotic curve is maintained. However, there are several differences. We measured the lordotic angle in various positions (standing, supine and during axial traction) and areas (each vertebral lordotic angle) by radiologic assessment. In addition, we also acquired the distances of anterior and posterior intervertebral space and could calculated the A/P ratio and compare it in various conditions. The A/P ratios of the LCCT group were similar to the initial state in all vertebral segments but not in the ST group. Interestingly, the A/P ratio of the L3/4 segment in the LCCT group, where the previous report [13] gave an anterior translation on L4, was not significantly different than the other groups in our study (1.46, 1.43, and 1.44, in initial, LCCT, and ST group, respectively). Also, the A/P ratio of LCCT was even more decreased than the initial group in the L5/S1 segment. As during LCCT, unnecessary pressure was not given to the posterior structures, the mean increases of intervertebral spaces in all segments were also higher than other groups: L3/4 were 23.62, 27.52, and 26.04 mm, and for L4/5 were 26.26, 29.82, and 27.49 mm, in initial, LCCT, and ST groups, respectively (Figure 2). It means that LCCT increases the traction distance more than ST.

There was also a similar study to increase the lordotic angle in 2002 [16]. By applying 3-point bending lumbar extension traction, lumbar lordotic curve could be maintained. However, the study was only applied to the chronic LBP subjects with hypolordosis and did not measure the changes quantitatively. Simply measuring the whole intervertebral angle in the previous reports [13,16] might not be enough to evaluate the precise segmental and geometrical differences during the new traction method. In this sense, our study could be a first study to evaluate the angular and distance changes in all directions and levels using LCCT quantitatively.

In the present study, anterior and posterior intervertebral distance ratios were greater in the supine position than during LCCT, which could also explain increased back pain in the supine position, as in this position, the lordotic curve decreases due to the influence of body weight. In previous studies, the average lordotic curve was reported to be 10.9° at the L4/5 intervertebral level [15,17]. In the present study, total angles (between the lower endplate of L2 and upper endplate of L5) and segmental angles (between L4 and L5, the structure most vulnerable to lumbar intervertebral disc disease) showed similar results. During LCCT, measured angles, distances, and distance ratios showed that lordotic curves were maintained in the supine position, and no participant reported discomfort or pain during treatment.

Several limitations of the present study warrant consideration. First, the study cohort was sufficient for our purposes, a larger cohort including patients with different demographics would have allowed us to determine the effects of age, weight, race, and gender. Second, we did not investigate the long-time treatment efficacy of low back pain patients, but rather geometrical changes caused by traction, and thus, real intervertebral disc compositional changes during LCCT were not measured. Thus, we observed the mechanical real-time response of spine only in patients without significant musculoskeletal abnormalities and the effect on patients with low back pain has not yet been studied. However, long-term clinical results could be different according to the patients’ characteristics. Further, Magnetic resonance imaging study is required to measure the intervertebral disc compositional changes caused by traction. Nevertheless, we believe our results are meaningful at a basic research level as they provide real-time data of geometric disc changes during LCCT in a clinical setting.

## 6. Conclusions

Based on radiologically measured dimensional changes, the developed lordotic curve-controlled traction device appears to be clinically useful for increasing intervertebral disc space evenly under traction by the maintaining lordotic curve.

## Figures and Tables

**Figure 1 healthcare-09-00125-f001:**
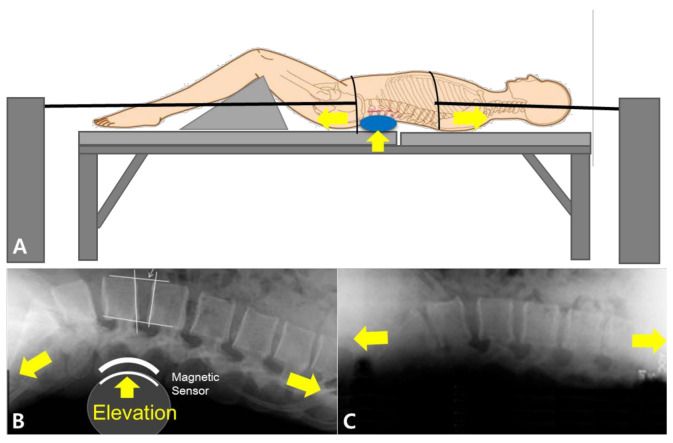
Lumbar traction techniques: (**A**,**B**), lumbar lordotic curve-controlled traction (LCCT) and (**C**), standard spinal traction (ST). The LCCT maintains the lordotic curve of the spine during traction by elevating the lordotic curve support.

**Figure 2 healthcare-09-00125-f002:**
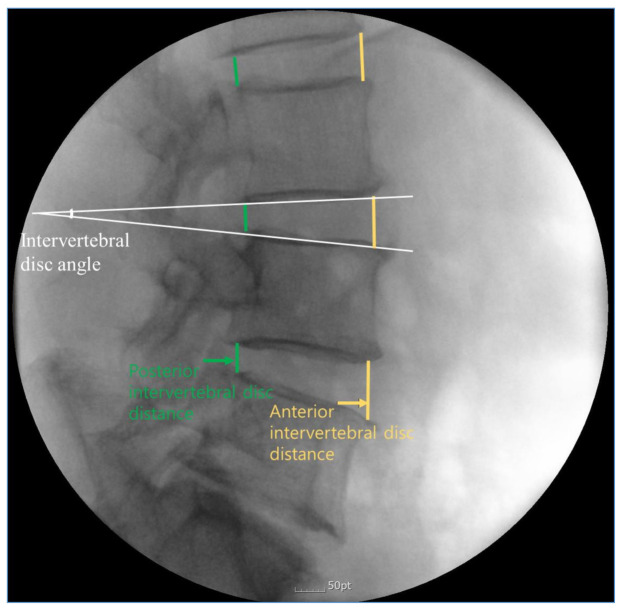
Illustration showing the measurement of intervertebral disc angle at L3/4 level, and intervertebral disc distance of anterior and posterior side.

**Figure 3 healthcare-09-00125-f003:**
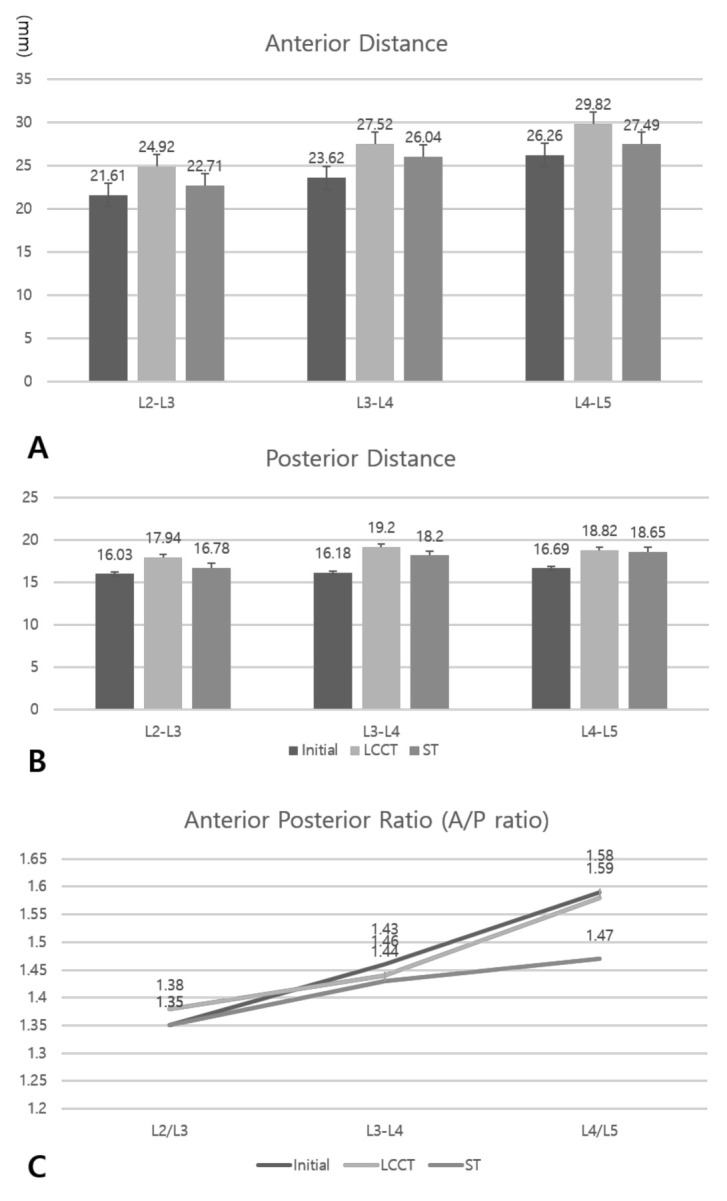
The intervertebral disc distances (**A**) in anterior and (**B**) posterior side in each vertebra. (**C**) The ratios of anterior and posterior distance in each vertebra. Abbreviation: initial indicates pre-operative position; LCCT, lordotic curve-controlled traction device; ST, standard traction.

**Figure 4 healthcare-09-00125-f004:**
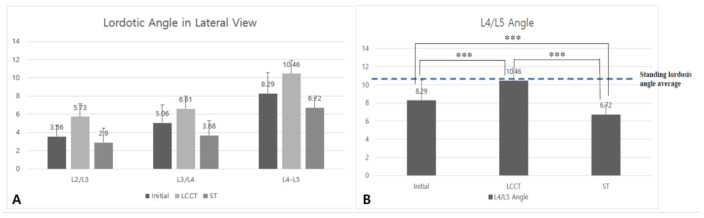
(**A**) The intervertebral disc angle in each vertebra. (**B**) In L4/5 level in three condition. Standing lordosis angle average is 10.9. Our patient in the initial group has the lowest L4/L5 angle and LCCT group has the highest L4/L5 angle. It has statistical significance among initial, LCCT, and ST (*p* = 0.000 ***).

**Table 1 healthcare-09-00125-t001:** General characteristics of the participants.

Variables	Male (n = 13)	Female (n = 27)
Age (years)	38.38 ± 10.53	41.22 ± 16.79
Height (cm)	171.84 ± 3.99	161.44 ± 3.28
Weight (kg)	75.30 ± 7.15	52.62 ± 4.86
BMI	23.66 ± 4.02	20.26 ± 2.29

All values are mean ± standard deviation. Abbreviation: BMI: Body mass index.

## Data Availability

The data presented in this study are available in insert article.

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
