# Peer review of "The Real Time Geometric Effect of a Lordotic Curve-Controlled Spinal Traction Device: A Randomized Cross Over Study"

_healthcare, 2021, doi:10.3390/healthcare9020125_

Round 1
Reviewer 1 Report
Chang-Hyung Lee and coauthors have analyzed the clinical efficacy of a lordotic curve controlled spinal traction (LCCT) device by comparing it with standard spinal traction (ST) device that was designed to straighten the spine without considering physiological lumbar lordosis. The authors suggest that Lordotic Curve Controlled Traction (LCCT) maintains the lordotic curve during traction, thus the traction force would be applied to the posterior spinal structure effectively. However, the reviewer thinks that the manuscript needs more detailed information about their experimental method and result description to help the readers to better understand. In addition, the criteria of the patient seem not to be determined well. Therefore, the reviewer thinks that several revisions are needed before publishing this manuscript in Healthcare
- The author selected 40 patients who have mild non-radicular low back pain. However, the cause of the pain was not clearly classified. The reviewer wonders that the efficacy would be different or show any tendency depending on the cause of the symptom.
- The devices used in this experiment are not well described. It would help the readers understand if the authors show how the devices look like, how they are applied to the patient. Besides, the reviewer wonders if the devices are commercial products or custom-designed devices.
- The reviewer thinks that it is more reliable if the author can show the representative spinal images from both the LCCT group and the ST group.
- Figure 1and Figure 2 need detailed descriptions. The main manuscript tells the figure 1 summarized LCCT and ST traction method, but Figure caption only said LCCT device. Besides, it is hard to understand what the image, arrow, and lines tell the readers. Each mark needs an explanation that tells what it means. It would be helpful if Figure 2 also include the description that where should the reader look for intervertebral distance and distance ratio.
- There is a grammar error in the Figure 2 caption.
- Each figure in Figure3 and 4 need to be included in the main manuscript separately.
- The value of data in the Figure 3A does not match with the value of the data in manuscript
Author Response
Thank you for the detailed questions and kind comment on the manuscript. We fully reviewed the questions. We answered and corrected the unclear sentences.
Please see the attachment.
Reviewer 1:
- The author selected 40 patients who have mild non-radicular low back pain. However, the cause of the pain was not clearly classified. The reviewer wonders that the efficacy would be different or show any tendency depending on the cause of the symptom.
- Unless there are significant spinal asymmetry or deformity, there could be no apparent difference according to the patient’s cause while applying real time traction. In this study, we observed the ‘mechanical response’ of spine only in patients without significant musculoskeletal abnormalities. However, long term ‘clinical results’ could be different according the patient’s characteristics as you suggested. We mentioned this in the discussion. (Line 235-237)
- The devices used in this experiment are not well described. It would help the readers understand if the authors show how the devices look like, how they are applied to the patient. Besides, the reviewer wonders if the devices are commercial products or custom-designed devices.
- We added the more detailed descriptions of the device used(figure 1). The main difference of this device is that it maintains the lordotic curve of the spine during traction by elevating the lordotic curve support. We used a commercial product ( Kinetrac-9900, Hanmed Co., Gimhae, Korea) with some modifications to set the clinical setting. We mentioned this in the manusciption. (Line 88-90)
- The reviewer thinks that it is more reliable if the author can show the representative spinal images from both the LCCT group and the ST group.
- We added the pictures to clarify the method (figure 1)
- Figure 1and Figure 2 need detailed descriptions. The main manuscript tells the figure 1 summarized LCCT and ST traction method, but Figure caption only said LCCT device. Besides, it is hard to understand what the image, arrow, and lines tell the readers. Each mark needs an explanation that tells what it means. It would be helpful if Figure 2 also include the description that where should the reader look for intervertebral distance and distance ratio.
- We changed the figure 1 and descriptions as follows.
Figure 1. Lumbar traction techniques: A and B, lumbar lordotic curve-controlled traction (LCCT) and C, standard spinal traction(ST). The LCCT maintains the lordotic curve of the spine during traction by elevating the lordotic curve support.
- There is a grammar error in the Figure 2 caption.
- We changed the figure 2 and descriptions as follows.
Figure 2. Illustration showing the measurement of intervertebral disc angle at L3/4 level, and intervertebral disc distance of anterior and posterior side.
- Each figure in Figure3 and 4 need to be included in the main manuscript separately.
- Thank you for the kind comment. We separated the figure 3 and 4.
- The value of data in the Figure 3A does not match with the value of the data in manuscript
- Thank you for the kind comment. An error occurred when the unit was changed from PT to mm. All data has been modified in mm.(figure3A, B)
Figure 3. The intervertebral disc distances (A) in anterior and (B)posterior side in each vertebra. (C) The ratios of anterior and posterior distance in each vertebra.
Abbreviation: initial indicated pre operative position; LCCT, lordotic curve controlled traction device ; ST , standard traction

Reviewer 2 Report
thank you for your article.
Nevertheless I see a lot of shortcomings which contradict publication in HealthCare:
No guideline on low back pain recommends traction for mnaging chronic low back pain. There is no evidence. Furthermore any rational paradigm is missing because there is no association between age related disc narrowing and low back pain.
Despite this lacking rationale all participants - after randomization not clearly described - were treated under radition. Radiation can harm the participants health.
Furthermore there is no rational paradigm that traction will relief lbp because morphological changes to the aging lumbar discs will continue.
furthermore aging of the discs and disc narrowing does not predict low back pain so working on this issue does not interfere the biopsychsocial origin of chronic low back pain.
Author Response
Thank you for the detailed questions and kind comment on the manuscript. We fully reviewed the questions. We answered and corrected the unclear sentences. Please see the attachment
- Thank you for the comments. As you suggested, there were mixed results regarding the effect of traction. We also stated the previous mixed results in the introduction (line 38~44). Although randomized, well designed reports were lacking, traction has been suggested widely and steadily in clinics due to its theoretical benefits in relieving mechanical disc space compression. Likewise, various physical modalities and exercises are permitted and widely used in spite of lack of well- designed controlled study.
Due to its theoretical benefits in that elongating the compressed intervertebral space, traction has been widely used in clinical settings. Thus the authors tried to standardize the treatment protocols according to the ‘theoretically safe’ guidelines. In this sense, we tried not to exceed the ‘natural lordotic spinal curve’ during traction rather maintaining the ‘original curve’. If no excess curve deterioration occurs but still elongate the compressed intervertebral disc space, there could be no harm at all. In addition, we only apply this method once while monitoring the patient’s discomfort. IRB, patient’s consent, and doctor’s monitoring were also given during the whole procedures to avoid any complication. - Also, there are many previous reports regarding the occurrence of degeneration according to age. As you suggested, ‘morphological changes will continue’ as one gets age.
The common source of low back pain consists of mechanical (nerve entrapment due to disc degeneration) and chemical (inflammation or irritation surrounding nociceptive structure) factors. The clear source of low back pain could not be identified just by relating the narrow spinal structure and pain (as there has been poor relation between mechanical compression and pain in previous studies).
In this study, we only evaluated the ‘real time response of traction’ by comparing the lordotic curve controlled traction device and the standard, traditional traction method.
it does not mean that LCCT could solve all the low back pain rather it could at least decrease the severity of the ‘mechanical compression’.
Regarding the safety problems, we took radiographs before and during traction which does not exceed longer than other spinal intervention procedures.
In general, traction treatment is usually performed in our clinics to relieve disc space narrowing regardless of traction type (ST and LCCT). Also, traction treatment should be equipped in training hospital (mandatory) and widely used in Korea due to its theoretical benefits. Also, IRB and consent form were given to all the patients recruited.

Reviewer 3 Report
The article presents the efficacy of a lordotic curve spinal traction device.
Strength points of the article:
- proposal of a new device used for traction of mild, non-radicular low back pain
- maintainance of the lordodic curve during the traction
Weakness:
- the randomization procedure not sufficiently described
It is stated in the title that the study is randomized. However, the randomization process is not presented in the Methods section. If the study includes a new device, is it registered in a clinical trial register?
Why were the patients with low back pain symptoms less than 3 months excluded?
Figure 1 presents a X-ray image. A figure of the new device should also be included.
There are several phrasing mistakes (see lines 48, 69).
Author Response
Thank you for the detailed questions and kind comment on the manuscript. We fully reviewed the questions. We answered and corrected the unclear sentences.
Please see the attachment.
It is stated in the title that the study is randomized. However, the randomization process is not presented in the Methods section.
- Thank you for the kind comment.
We recruited the patients in random order with cross over study design.
- To clarify the method, we added the necessary part. (Line 106-110)
- The patients were taken traction method ST or LCCT in random order. The random allocation was decided by tossing a coin. When the head side is come out, the subjects was assigned to ST treatment first and the other, adversely. The first intervention was applied for 10 min and subjects had a washout period of 20 min between each intervention. After this, the patients were taken other traction in cross over study design
If the study includes a new device, is it registered in a clinical trial register?
- Thank you for comments. We added the more detailed descriptions of the device used as follows. . (Line 82-84)(Line 86-88)
- We used a commercial product ( Kinetrac-9900, Hanmed Co., Gimhae, Korea) with some modifications to set the clinical setting.
- We registered the device and the study was approved by the local and the central ethical committees at our hospital (03-2016-008).
Why were the patients with low back pain symptoms less than 3 months excluded?
- During the initial, acute stage of low back pain, muscle guarding or spasm occurs which exacerbate the spinal asymmetry and nerve root irritation. Thus unnecessary muscle guarding could interfere the real time effect of traction in this study. We mentioned this in the methods(Line 80-82)
There are several phrasing mistakes (see lines 48, 69).
- We change the phrase as follows;
- Spinal traction (ST) is a means of spinal decompression and was designed to decompress straighten the spine to improve spinal alignment.
- à Spinal traction (ST) is a form of decompression therapy which was designed to relieve pressure on the spine and improve spinal alignment.
- In the supine position, the lordotic curve decreases due to vertical pressure on the lumbar curve, which is severe in patients with low back pain [12]. When traction force is applied to the spine in the supine position, it primarily straightens the natural lordotic curve without unduly decompressing the intervertebral discs, and thus, rather than decompressing the intervertebral discs, ST compromises the lordotic curve [13].
- ->In the supine position, the lordotic curve decreases due to vertical pressure on the lumbar spine, which is more decreased in patients with low back pain [12]. When traction force is applied in the supine position, it primarily straightens the natural lordotic curve without unduly decompressing the intervertebral disks. Thus, rather than decompressing the intervertebral discs, traditional traction decreases the lordotic curve
Figure 1 presents a X-ray image. A figure of the new device should also be included.
- Thank you for your comments. The figure of new traction device was added in the manuscript as follows.
- Figure 1. Lumbar traction techniques: A and B, lumbar lordotic curve-controlled traction (LCCT) and C, standard spinal traction(ST). The LCCT maintains the lordotic curve of the spine during traction by elevating the lordotic curve support.

Round 2
Reviewer 2 Report
Thank you for your revisions.
After all my concerns to quality and impact of this article are not dispelled.
Author Response
Thank you for the detailed questions and kind comment on the manuscript. We fully reviewed the questions. We answered and corrected the unclear sentences. Reviewer 2 thank you for your article. Nevertheless I see a lot of shortcomings which contradict publication in HealthCare: No guideline on low back pain recommends traction for mnaging chronic low back pain. There is no evidence. Furthermore any rational paradigm is missing because there is no association between age related disc narrowing and low back pain. Despite this lacking rationale all participants - after randomization not clearly described - were treated under radition. Radiation can harm the participants health. Furthermore there is no rational paradigm that traction will relief lbp because morphological changes to the aging lumbar discs will continue. furthermore aging of the discs and disc narrowing does not predict low back pain so working on this issue does not interfere the biopsychsocial origin of chronic low back pain. Thank you for the comments. As you suggested, there were mixed results regarding the effect of traction. We also stated the previous mixed results in the introduction (line 51~57). Although randomized, well designed reports were lacking, traction has been suggested widely and steadily in clinics due to its theoretical benefits in relieving mechanical disc space compression. Likewise, various physical modalities and exercises are permitted and widely used in spite of lack of well- designed controlled study. Due to its theoretical benefits in that elongating the compressed intervertebral space, traction has been widely used in clinical settings. Thus the authors tried to standardize the treatment protocols according to the ‘theoretically safe’ guidelines. In this sense, we tried not to exceed the ‘natural lordotic spinal curve’ during traction rather maintaining the ‘original curve’. If no excess curve deterioration occurs but still elongate the compressed intervertebral disc space, there could be no harm at all. In addition, we only apply this method once while monitoring the patient’s discomfort. IRB, patient’s consent, and doctor’s monitoring were also given during the whole procedures to avoid any complication. Also, there are many previous reports regarding the occurrence of degeneration according to age. As you suggested, ‘morphological changes will continue’ as one gets age. The common source of low back pain consists of mechanical (nerve entrapment due to disc degeneration) and chemical (inflammation or irritation surrounding nociceptive structure) factors. The clear source of low back pain could not be identified just by relating the narrow spinal structure and pain (as there has been poor relation between mechanical compression and pain in previous studies). In this study, we only evaluated the ‘real time response of traction’ by comparing the lordotic curve controlled traction device and the standard, traditional traction method. it does not mean that LCCT could solve all the low back pain rather it could at least decrease the severity of the ‘mechanical compression’. Regarding the safety problems, we took radiographs before and during traction which does not exceed longer than other spinal intervention procedures. In general, traction treatment is usually performed in our clinics to relieve disc space narrowing regardless of traction type (ST and LCCT). Also, traction treatment should be equipped in training hospital (mandatory) and widely used in Korea due to its theoretical benefits. Also, IRB and consent form were given to all the patients recruited.

Reviewer 3 Report
The authors addressed my comments except for the registration of the study. Considering the fact that a new device was used the study must be registered in a database. This does not refer to the hospital ethical committee.
Author Response
We used a commercial product ( Kinetrac-9900, Hanmed Co., Gimhae, Korea) that was slightly modified version of the original type of traction
but still remains the original traction mechanism with the support of the manufacturer to set the clinical setting(only to be adjusted for c-arm).
We mentioned this in the manusciption. (Line 88-90). And we approved the hospital ethical committee of this device